# Vasopressin as a Possible Link between Sleep-Disturbances and Memory Problems

**DOI:** 10.3390/ijms232415467

**Published:** 2022-12-07

**Authors:** Bibiána Török, János Varga, Dóra Zelena

**Affiliations:** 1Center for Neuroscience, Szentágothai Research Center, Institute of Physiology, Medical School, University of Pécs, 7624 Pécs, Hungary; 2Laboratory of Behavioral and Stress Studies, Institute of Experimental Medicine, 1183 Budapest, Hungary

**Keywords:** vasopressin, circadian rhythm, sleep, EEG, memory, novel object recognition, Brattleboro rat

## Abstract

Normal biological rhythms, including sleep, are very important for a healthy life and their disturbance may induce—among other issues—memory impairment, which is a key problem of many psychiatric pathologies. The major brain center of circadian regulation is the suprachiasmatic nucleus, and vasopressin (AVP), which is one of its main neurotransmitters, also plays a key role in memory formation. In this review paper, we aimed to summarize our knowledge on the vasopressinergic connection between sleep and memory with the help of the AVP-deficient Brattleboro rat strain. These animals have EEG disturbances with reduced sleep and impaired memory-boosting theta oscillation and show memory impairment in parallel. Based upon human and animal data measuring AVP levels, haplotypes, and the administration of AVP or its agonist or antagonist via different routes (subcutaneous, intraperitoneal, intracerebroventricular, or intranasal), V1a receptors (especially of hippocampal origin) were implicated in the sleep-memory interaction. All in all, the presented data confirm the possible connective role of AVP between biological rhythms and memory formation, thus, supporting the importance of AVP in several psychopathological conditions.

## 1. Introduction

Sleep and memory disturbances are common in many psychiatric disorders (e.g., schizophrenia [1], autism [2,3], and post-traumatic stress disorder [4]). A better understanding of the mechanism of these symptoms might be important for treatment improvement. Vasopressin (AVP)—in addition to playing an important role in the regulation of salt-water homeostasis [5] and stress adaptation [6]—has also been implicated in sleep regulation [6,7,8,9]: the vasopressinergic cells of the suprachiasmatic nucleus (SCN) play an important role in determining the circadian rhythm [10]. Furthermore, AVP also seems to be important for memory formation [11,12].

To widely explore the mechanisms of basic functions, such as sleep, animal models are essential. Knocking out a gene might shed new light on its physiological role. Therefore, the vasopressin-deficient Brattleboro rat (homozygous recessive for diabetes insipidus (di/di)) may be suitable for studying the vasopressinergic regulation of sleep and memory formation with the hope of confirming a connection between these two processes. Indeed, in this genetically AVP-deficient rat strain sleep disturbances have already been described [13] and several research groups have already reported memory impairment [14,15,16]. The importance of this strain is further confirmed by the fact that AVP-deficient mice are not viable, which suggests that in the Brattleboro rat the negative consequences of peripheral AVP deficiency (e.g., hypotension and water loss) are well compensated [17].

The aim of this review was to summarize the information found in the literature on the role of AVP in the regulation of circadian rhythmicity (including sleep) and try to connect it with its role in memory formation. Taking into consideration the above-mentioned value of the AVP-deficient Brattleboro rat, we have focused on this strain.

## 2. Vasopressin

AVP (vasotocin in birds [18] and lypressin, which contains lysine instead of arginine, in pigs [19]) is a nonapeptide. As a hormone, it is released into the general circulation from the posterior pituitary and forms the peripheral vasopressinergic system, which is also called the hypothalamic–neurohypophyseal system [20]. This is the main regulator of water resorption in the kidney and plays an important role in salt-water homeostasis (hence it is also known as the antidiuretic hormone) [21]. However, AVP, released within the brain, constitutes a central vasopressinergic system and regulates the activity of other neurons. This latter system is involved in circadian sleep-wake regulation [22,23], anxiety [24], depression [25], learning and memory (see later), as well as social behavior [26].

### 2.1. Receptor Subtypes

AVP may target three different receptor types. The first two are the V1a (vascular) and V1b (hypophyseal) receptors, which utilize the Gq-phospholipase C pathway, while V2 (kidney) operates through the Gs-adenylate cyclase stimulating pathway.

The V2 receptor (V2R) is the predominant form in the kidney and regulates salt-water homeostasis, while V1bR, which can be found primarily on the anterior lobe of the pituitary, is implicated in hypothalamic-pituitary-adrenocortical (HPA) axis (also known as the stress axis) regulation. At the periphery, V1aR can be found on vessels that are responsible for the vasoconstrictor effect of AVP (hence the name vasopressin).

However, V1aR is also the dominant form in the central nervous system and is associated with various behavioral and cognitive effects [27]. In most species, this receptor subtype can be found in the amygdala, bed nucleus of stria terminalis (BNST), lateral septum (LS), hypothalamus, and brainstem (Figure 1A). In Rhesus monkeys, both V1aR mRNA (in situ hybridization) and ligand binding (autoradiography) were detected on several cortical sites including the prefrontal cortex (PFC), cingulate, pyriform, and entorhinal cortex, as well as presubiculum and mamillary bodies [27]. Further studies also confirmed the wide distribution of V1aR in rat cortical areas [28].

### 2.2. Anatomy

The vast majority of AVP is synthesized in two magnocellular cell groups: the supraoptic nucleus and the paraventricular nucleus of the hypothalamus (PVN) [30] (Figure 1B). The magnocellular neurons of these nuclei send their axons to the posterior pituitary, where the neuropeptide is stored and released into the general circulation and acts as a hormone.

However, the same neurons might secrete AVP from their soma and dendrites to the extracellular fluid [31] called ‘intranuclear release’ or ‘intra-hypothalamic release’ [32,33]. This intranuclearly released neuropeptide might reach adjacent areas (e.g., the LS or medial amygdala (MeA)) through volume transmission in physiologically relevant concentrations and regulate neuronal activity, thus, leading to—among other things—altered behavior [34,35,36,37].

Parvocellular neurons also synthesize AVP (Figure 1C). In the medial part of the PVN, AVP co-localized with a corticotropin-releasing hormone, and they reached the median eminence and long portal vessels together and can interact on the anterior pituitary to regulate adrenocorticotropic hormone secretion (the hypophyseal component of the HPA axis), thereby participating in stress regulation [38]. Another brain area responsible for synthesizing AVP is the amygdala, especially MeA [39]. This area is important for regulating social behavior. Human studies have suggested that AVP modulates medial PFC-amygdala circuitry during emotion processing [40]. AVP-producing neurons (not only the V1aR mentioned before) can also be found in BNST and LS.

## 3. The Brattleboro Rat as a Model of AVP-Deficiency

The Brattleboro rat strain was discovered in 1961 in Brattleboro, Vermont. It evolved from the Long–Evans rat strain through a random autosomal recessive mutation (Figure 2A). The genetic mutation of the Brattleboro strain was later confirmed to be a single nucleotide deletion of a G residue in the second exon of the neurophysin gene. Therefore, a reading frame shift develops, which results in a different C-terminus for the precursor hormone (Figure 2B). Due to the lack of a stop codon, the mRNA cannot leave the ribosomes after translation, and the incomplete protein chain possibly sticks to the ribosome. Consequently, a ubiquitin ligase (LTN1), which is part of the ribosome-associated quality control complex, induces proteolysis. All in all, biologically active AVP will not be present in the central nervous system of the Brattleboro rat. It has to be emphasized that at the peripheral organs (e.g., the colon, liver, kidney, and testis), AVP may be produced in a different way; therefore, Brattleboro rats are not wholly AVP-knockout (KO) animals. However, their blood lacks AVP (which is produced in magnocellular, neurosecretory hypothalamic cells); therefore, they develop central diabetes insipidus with polydipsia and polyuria [41,42,43,44,45] (Figure 2C).

According to the literature, the Brattleboro rat is a good model for investigating central AVP effects. This strain shows memory and sleep impairments [14,46]; therefore, we can get closer to understanding the role of AVP in regulating circadian rhythms and the connection between sleep and memory by studying di/di rats.

## 4. Circadian Rhythm

Circadian rhythms occur in various endogenous processes with a cycle of about 24 h. These rhythmical changes are observable mostly in the behavior and physiology of the individuals [47].

### The Circadian Rhythms of Brattleboro Rats

Our previous studies did not find circadian disturbances in the AVP-deficient Brattleboro rats based on their locomotion [15]. Interestingly, other authors reported hyper-locomotion in adults [48,49] and pups [50,51] and hypo-locomotion in adolescent rats [52], but our previous studies on adult animals detected no locomotor abnormalities during short (up to 15 min) behavioral tests (males [14,53,54] and females [55,56]) as well as during a 24 h constant recording [15]. One reason for the contradictory results could be that different authors used different control animals. The Brattleboro strain was developed from a Long–Evans colony in 1961, and since then, the two strains may have diverged genetically (Figure 2A). Firstly, we compared heterozygous littermates [53] because the effect of maternal influence can be balanced this way (see the effect of AVP on maternal behavior in [55], and see the effect of maternal neglect on locomotion in [57]). However, we subsequently breed out a +/+ line, which has no genetic deficits but has a close relationship with the di/di animals (see [58] for the breeding details), and we compared di/di rats with +/+ rats [14,16,55,56]. Nevertheless, we could not detect any locomotor differences in any of the cases and that were independent of the rat’s sex. Another possible explanation for the discrepancies might be related to the observation time. Most of the studies conducted experiments during the light period of the day [52], while we conducted our examination in the dark, active phase of the animals [53].

## 5. Suprachiasmatic Nucleus (SCN), the Endogenous Clock

Approximately fifty years ago, for the first time, lesion experiments demonstrated the important role of the suprachiasmatic nucleus (SCN) in the regulation of circadian rhythms [59]: its lesion in rats caused circadian rhythm loss in drinking and locomotor behavior (Figure 1D). Further SCN extirpation experiments on rats showed the elimination of the circadian rhythm of corticosterone, which is the major end-hormone responsible for stress adaptation [60]. Later studies in cell cultures confirmed that SCN neurons have autonomous circadian firing rhythms [61]. Since then, several studies supported the key role of the SCN in the regulation of circadian rhythms [62,63].

However, subsequent experiments discovered the presence of endogenous clocks in other parts of the brain [64,65] as well as in other organs (e.g., the gut [66]). Thus, the SCN seems to be the central coordinator of numerous autonomous cellular clocks.

Nevertheless, the first neurotransmitter that was discovered in the SCN was AVP, which was also proven to influence circadian rhythms [62]. Although disturbances of the endogenous clock are more typical of depression, it was also implicated in schizophrenia [67]. Its importance is supported by the jet lag-induced exacerbation of psychosis and seasonal changes in the prevalence of psychotic episodes. Moreover, AVP was implicated in both depression [53,56] and schizophrenia [14,15,68].

### 5.1. Suprachiasmatic Nucleus and Vasopressin

The presence of AVP as well as its V1aR and V1bR were confirmed in the SCN of many species, including humans [22,69]. In cats, the concentrations of AVP in the cerebrospinal fluid (but not in the plasma) were approximately five times higher during morning hours than during night hours [70], while in mice, which are nocturnal animals, the V1aR mRNA level of the SCN peaked during night hours [71]. Thus, it can be hypothesized that the release of AVP from the SCN indicates the switch to the resting phase with down-regulation of the HPA axis, reproduction, and locomotor activity [72]. Indeed, the vasopressinergic output of the SCN innervates brain areas that are important for the regulation of hormone secretion and behavior [73]. Interestingly, AVP is not only a major output but also a retinal vasopressinergic innervation of the SCN, though V1aR might also be important for the regulation of the circadian clock [74].

In accordance with the important regulatory role of AVP, the activity period, precision, and phase relationships of SCN neurons were altered in an AVP-Cre mice model with lower AVP levels [75]. In V1aR KO mice, the circadian rhythmicity of locomotor activities was also significantly reduced [71], and the V1aR and V1bR double KO animals adapted faster to circadian rhythm changes (i.e., jet-leg simulation) than their wild-type littermates [76,77]. In agreement, the pharmacological blockade of the V1aR and V1bR in the SCN resulted in accelerated recovery from jet lag [76]. It was also suggested that AVP, but not vasoactive intestinal peptide (VIP)-containing neurons (another important neuropeptide of the SCN), is essential for the autonomous network synchrony of the SCN (using Cre-mice and color-switch-luciferase reporter proteins) [78].

However, other studies confirmed the significant role of VIPs in the regulation of circadian behavior [79]. Its levels showed circadian variation even in the human plasma [80]. In accordance with this, in both the V1aR KO [71] and the V1aR-V1bR double KO mice [76], the oscillation of the clock genes remained normal. Thus, we can conclude that AVP and VIPs work to regulate different aspects of the circadian clock in the SCN [81,82,83] or act during different developmental stages [84]. Other neuropeptides, such as pituitary adenylate cyclase-activating polypeptides from the retina, may also have a great impact [82].

### 5.2. The Suprachiasmatic Nucleus of the Brattleboro Rat

The peak of spontaneous neuronal activity in the SCN displayed during the subjective light phase (the passive sleep phase of rodents) was significantly lower in the AVP-deficient Brattleboro animals that had otherwise no disturbances in the circadian variation of their electric activity [85]. This suggests AVP’s possible rhythm-amplifier role. In line with this mild disturbance in the circadian clock, Brattleboro rats displayed less depression-like behavior [53,56] and more schizophrenia-like [14,15,68] behavior in both sexes.

All in all, these results highlight the potential of AVP for the management of circadian rhythm misalignments, such as jet lag, shift work, and other psychiatric disturbances.

## 6. Importance of Sleep in Physiology and Disease

One of the most important biological rhythms is sleep, which is essential for health [86]. Living organisms sleep to recover. Sleep is homeostatically regulated in all animal species studied so far [87]. Irregular sleep or insomnia can induce various diseases including cardiovascular disease, cancer, and psychiatric disorders [88,89]. Sleep can also influence the immune system [90]: slow-wave sleep promotes memory consolidation not only in the brain but also in the immune system. For example, sleep deprivation affected T-helper cell activity in a self-controlled human study [91]. Furthermore, sleep disturbances can increase the risk of infectious diseases [92], and might influence mortality; however, the causality is not proven [93].

Many studies suggested that sleep has an impact on memory consolidation (for reviews and meta-analyses see [94,95,96,97,98]). Insomnia (or circadian rhythm disturbances) can deteriorate memory formation and consolidation [99,100,101]. As memory problems are key symptoms of many disorders, it is not surprising that sleep disorders may be related to the appearance of neuropsychiatric symptoms in psychosis, neurodegenerative diseases, major depressive disorder, generalized anxiety disorder, post-traumatic stress disorder, and bipolar disorder (DSM-V).

### 6.1. Electroencephalographic (EEG) Stages of Sleep

Sleep phases can be characterized based on typical EEG power as well as muscle activity (for the determination of the rapid eye movement (REM) phase). Based on the EEG register, stages of sleep can be determined and separated [102]. In humans, during a calm, alert state with closed eyes, alpha waves with a frequency of 8–13 Hz can be registered on the EEG. When the eyes are opened, this is replaced by an even faster wave, the beta rhythm (14–30 Hz), which means quick potentials with small amplitudes typical of the awake, attentive state.

In the first stage of sleep (S1), the alpha waves gradually disappear, and theta waves appear on the EEG instead. In the second phase (S2), during superficial sleep, sleep spindles and K-complexes are visible. Superficial sleep deepens, theta waves (4–7 Hz) become more frequent and slower, and lower-frequency delta waves (1–4 Hz) appear. During the third stage (S3), sleep deepens even further. Instead of sleep spindles and K-complexes, the EEG register is characterized by slow delta (<3.5 Hz) waves. Because of the slow EEG waves characteristic of the second and third phases, we collectively call them slow-wave sleep (SWS or NREM). The fourth stage is deep sleep, or REM (rapid wave sleep or paradoxical sleep). During this phase, EEG is characterized by the absence of sleep spindles and K-complexes, and instead of the slow delta wave, the faster theta wave appears. Rapid eye movements can be detected during this phase. Due to the faster theta waves and rapid eye movements, this stage is closer to the awake state and first stage of sleep, and due to the complete cessation of muscle tone, this stage is closer to deep sleep; therefore, it is called paradoxical sleep [102].

The best-characterized marker of sleep homeostasis is NREM sleep (EEG power between 0.5 and 4 Hz), reflecting the accumulation of sleep pressure due to the prior wake state. It increases after spontaneous awakenings and short-term (3–24 h) sleep deprivation but decreases during sleep.

### 6.2. Theta Waves

Theta (4–7 Hz) waves or oscillation—especially detectable in the hippocampus during active behavior [103] as well as during the REM phase [104]—are highly connected to learning and memory formation [105,106]. In mice, lipopolysaccharide- (a Gram-negative bacterium wall) induced fear memory decline was—at least partly—mediated by a decrease in theta oscillation [107]. Theta-like activity may occur in close association with visual exploration as it was detected in rodents during running, sniffing, and whisking [108]. During learning, the hippocampus goes from a “random” to a more “organized” state with the help of plastic changes such as the induction of long-term potentiation. These plastic changes occur preferably during particular phases of theta. On the other hand, theta oscillations recorded over the frontal cortices (which is also called the frontal theta) are associated with language processing in humans [109] and are a key source of top-down control of cognitive processes in general [110]. In accordance with this idea, increased theta power was found for later remembered items versus later forgotten items during a human associative memory task [111]. Thus, the connection between theta oscillation and memory formation was confirmed not only in primates but also in humans [108].

### 6.3. Vasopressin and Sleep (Figure 3)

An early study on rats showed that intracerebroventricular (icv) administration of AVP increased wakefulness, and a general V1 receptor antagonist decreased it [112]. The sleep-influencing effect of AVP was also confirmed in humans [7]: systemic administration of AVP dose-dependently reduced S2 and REM sleep as well. This effect was independent of the HPA axis regulatory or vasopressor function of AVP. Therefore, it can be hypothesized that AVP regulates sleep via central, but not peripheral effects. In contrast, in a pilot study with 26 elderly subjects, chronic intranasal administration of AVP improved, rather than provoked, age-related sleep disturbance [8,9], and this was accompanied by a decrease in cortisol, the end-hormone of the HPA axis without any cardiovascular changes. These controversial results can be explained by a different regulation of AVP in the older age group. It is also possible that the chronic treatment has an opposite effect rather than an acute one (e.g., possible habituation to cardiovascular side effects).

**Figure 3 ijms-23-15467-f003:**
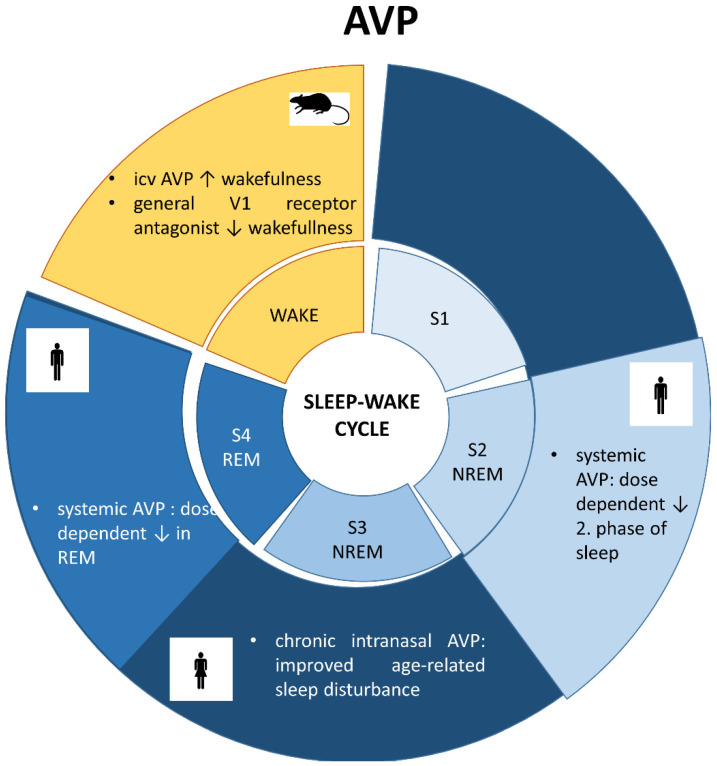
Sleep stages and involvement of vasopressin (AVP). S1: first stage of sleep, S2: second stage of sleep, S3: third stage of sleep; REM: radip eye movement; NREM: non-REM stage; icv: intracerebroventricular administration.

The AVP gene might be affected by epigenetic modification, which may contribute to its sleep-modifying effect. Namely, the hypomethylation of the AVP intron 1. was found in pregnant women who were previously exposed to psychosocial stress and had sleep disorders [113]. However, the causal link remains unclear.

### 6.4. Sleep in Brattleboro Rats

Few studies characterized the EEG of the Brattleboro animals. A detailed EEG analysis of the sleep phases [114] showed that during the subjective light period (the sleeping period of rodents), the di/di animals spent significantly more time awake. This shift towards the wake phase supports the sleep disturbance of these animals and confirms reduced circadian variability (see also [115]), which can be a sign of enhanced vulnerability [116]. Indeed, di/di rats spent less time in REM [46] and NREM sleep phases without much difference during their dark, active period. These differences were detectable both in the number of epochs as well as in the cumulative duration of the phases. In another experiment focusing on general pattern, smaller EEG wave amplitudes were measured during the REM and NREM stages of sleep comparing di/di to control animals [115].

### 6.5. Vasopressin and Theta Oscillation with Focus on Brattleboro Rat

In rats AVP administration—both icv as well as subcutaneously (sc)—reduced the power density, also known as “EEG slowing” [117]. Further studies focused on theta band (4–7 Hz) (Figure 4). A study with Wistar rats found that AVP administration might improve theta synchronization and coordination of theta-gamma coupling [118].

AVP-deficient animals showed reduced theta activity compared to their wild-type littermates, which was temporarily normalized by icv desglycinamide AVP administration (without a significant influence on polydipsia and polyuria) [13]. Therefore, we might conclude that AVP might be directly responsible for maintaining physiological theta activity, a key aspect of normal sleep and memory consolidation. This is supported by the icv AVP antiserum administration-induced theta activity decline towards the di/di level in the control animals [119]. The same research group revealed that an icv administration of an AVP analogue [119] (as well as intra-septal injections of AVP [120]) to di/di animals increased their mean theta frequency to the level of the wild-type animals. During the REM phase, however, only a tendentious difference (theta frequency of di/di: 6.4 cycles/sec vs. di/+: 6.8 cycles/second) was observed in the theta frequency range between AVP-deficient and control animals [121]. Nevertheless, the shift to lower frequencies (from 8 Hz to 6 Hz) was confirmed in our own colony as well.

All in all, the central vasopressinergic system might contribute to the normal functioning of the theta-rhythm-generating network.

## 7. Memory Disturbances Associated with Improper Sleep

### 7.1. Sleep-Dependent Memory Consolidation

Sleep deprivation studies widely confirmed the importance of sleep to normal human cognitive performance including learning and memory [122]. Even one night without sleep can diminish working memory performance [123]. Chronic restriction of sleep to 6 h or less per night produced cognitive performance deficits equivalent to up to 2 nights of total sleep deprivation [124]. Not only cognition but also the consolidation of motor memories is facilitated during sleep [125].

As previously mentioned (see Section 6.2.), hippocampal theta rhythm is associated with memory function [126,127] and memory consolidation [128]. During REM sleep, a regular theta rhythm occurs in the hippocampus [129]. However, NREM sleep is also associated with memory consolidation [130]. During SWS the memories are reactivated which helps to stabilize them [131].

Recent findings have challenged the sleep-memory hypothesis proposing that not sleep per se but sleep-induced plasticity is the determining factor of memory formation [132]. Indeed, the circadian system displays a remarkable degree of plasticity from cell to circuit-based levels [133], and plastic changes in vasopressinergic neurons were described in relation to salt-water homeostasis [134], rhythmic SCN activity [75], and memory-influencing hippocampal neurogenesis [135].

### 7.2. Vasopressin Function Related to Memory

It is well-accepted that AVP plays a role in learning and memory formation [11,12] (Figure 4). This role was first described in 1965 [136] and since then, it has been repeatedly confirmed. Similarly, enhanced SON AVP levels were implicated in the alleviation of sepsis-associated memory decline [137]. In male Wistar rats, acute sc AVP injections increased resistance to the extinction of pole-jumping-avoidance behavior in conditioned avoidance behavior tests [138]. A recent article found that repeated (but not acute) intranasal administration of AVP to prairie voles enhanced their spatial memory [139].

The hippocampus, a critical hub of spatial memory formation, seems to be a key point of AVP-mediated memory formation as well. In mice, AVP injections into the ventral hippocampus decreased, while antisera increased forgetting in a Go-No-Go visual discrimination task [140]. At a cellular level, AVP administration increased the in vitro electrical activity of the dentate gyrus cells (a gate that governs the inflow of information to the hippocampus) [141]. This effect was mediated by the V1aR, and its antagonist, SR49059, was able to prevent the AVP effect. Moreover, manipulating hippocampal V1bR had no influence on memory formation [142]. Human studies also confirmed the connection between the hippocampus (by grey matter volume and resting-state functional connectivity measured by magnetic resonance imaging), V1aR haplotypes, and verbal learning and memory performance evaluated by the California Verbal Learning Test-II [143]. Although there was no association between short-term verbal learning and memory and peripheral AVP levels in healthy individuals [144], this is not surprising when taking into consideration that peripheral AVP mainly comes from the magnocellular cells, while the brain AVP system mainly originates in the parvocellular cells (see Section 2.2). Indeed, peripheral administration of peptides might also lead to questionable results, leading to high, unphysiological concentrations that can subvert the interpretation of behavioral tests [145].

Interestingly, the inappropriate secretion of antidiuretic hormone syndrome, which is a state with elevated AVP levels, was also accompanied by impaired memory function (working memory impairment in T-maze) in V2 agonist-treated mice [146]. However, it might be connected to hyponatremia and not to the AVP level increase itself. In line with these contradictory results in a study using male Sprague-Dawley rats, intraperitoneal administration of a V1aR antagonist prevented trauma-induced cognitive decline detected by a Barnes maze test for assessing hippocampus-dependent spatial memory [147].

It was suggested that shorter metabolites, such as AVP (4–5)-NH_2_ dipeptide [148,149] or AVP(4-8) [135], might play a neuroprotective role via interacting with neurotrophins; however, their memory enhancing potential remains unelucidated.

In relation to EEG changes, the aforementioned enhancing effect of intra-septal injections of AVP on hippocampal theta rhythms [120]—a wave strongly associated with memory function [126,127] and memory consolidation [128]—suggests that AVP plays a direct role in regulating memory consolidation during sleep.

### 7.3. Memory Problems in Brattleboro Rats

As summed up above, AVP has a significant impact on memory formation. Therefore, it is not surprising that AVP-deficient Brattleboro animals were repeatedly reported to have memory deficits. In fact, this was one of their first behavioral abnormalities detected by active [150] or passive [151] avoidance tests, which are both highly aversive avoidance-based learning paradigms. The quicker extinction of the learned behavior of AVP-deficient animals was restored by subcutaneous AVP administration directly after the learning trial [145]. Although some later studies found contradictory results with similar aversive conditioning in di/di and control rats [152], this can be explained by the altered stress-adaptability of these animals [153,154].

Subsequent studies used the social recognition paradigm [68,155], and we confirmed social memory impairment in our local colony as well [14,15,16]. However, AVP has a substantial effect on social behavior independently of memory formation [156,157]; therefore, to study memory formation, the novel object recognition (NOR) test seems to be a better choice. Previously we described NOR disturbance in our di/di animals [15,16]. Donepezil, which is a cognitive enhancer, enhanced theta power during a NOR test [158]. In line with these observations, the memory deficit of the Brattleboro rat was accompanied by reduced theta power above the frontal cortical areas during both the sampling as well as the recall phase of the NOR. This suggests that AVP might be a possible link between memory formation and EEG disturbances.

## 8. Sex Differences

Interestingly, the memory decline of di/di rats was detectable in males but not in females [56]. Indeed, previous anatomical studies found a higher vasopressinergic fiber density in the brains of males than in the brains of females in rats [159,160,161], mice [73,162], prairie voles [163], and marsupials [164]. However, in some special brain areas, such as in the paraventricular nucleus of the thalamus, higher AVP immunoreactivity was detected in female, but not male rats [159].

Since the vasopressinergic SCN effect was sex-independent, there are likely sex differences in the downstream targets of the SCN [75]. However, no precise anatomical efferent target can be addressed (as SCN-derived diffusible signals reach the cerebrospinal fluid, forming a diffusible efferent pathway) [165]. Anyway, SCN neurons are part of the sympathetic and parasympathetic outflow [166] and may influence sleep and memory in this way as well as in a sex-dependent way because the autonomic function is sex-dependent [167].

## 9. Discussion

AVP expression is high in the SCN and shows circadian variation, which suggests its function in rhythmicity. Sleep and circadian rhythm disturbances are one of the leading symptoms of many neuropsychiatric diseases [89] and may lead to memory problems [99,100,101]. AVP might be a critical link between sleep and memory. Apart from human and animal data on AVP measurements, administration, and receptor manipulation, the AVP-deficient Brattleboro rats provided further evidence of this connection. Indeed, the disturbed memories of the Brattleboro rats were accompanied by disrupted circadian patterns with reduced sleep during the light, active phase [115] as well as reduced theta power [13], which was particularly significant during NOR.

All in all, we strengthen the hypothesis that central AVP plays a key role in regulating not only circadian rhythms, including sleep, but also memory and may serve as a link between the two processes by influencing sleep stages. V1aRs, especially in the hippocampus, were implicated in this connection.

## 10. Future Directions

Sleep and memory disturbances are very common symptoms of psychiatric disorders, and their treatment is not yet resolved. According to the current state of science, AVP can play an important role in the regulation of both processes, serving as a possible link between them. Thus, AVP treatment might boost the effectiveness of other therapies. Furthermore, AVP measurements might help diagnose disorders associated with sleep and memory impairments.

## Figures and Tables

**Figure 1 ijms-23-15467-f001:**
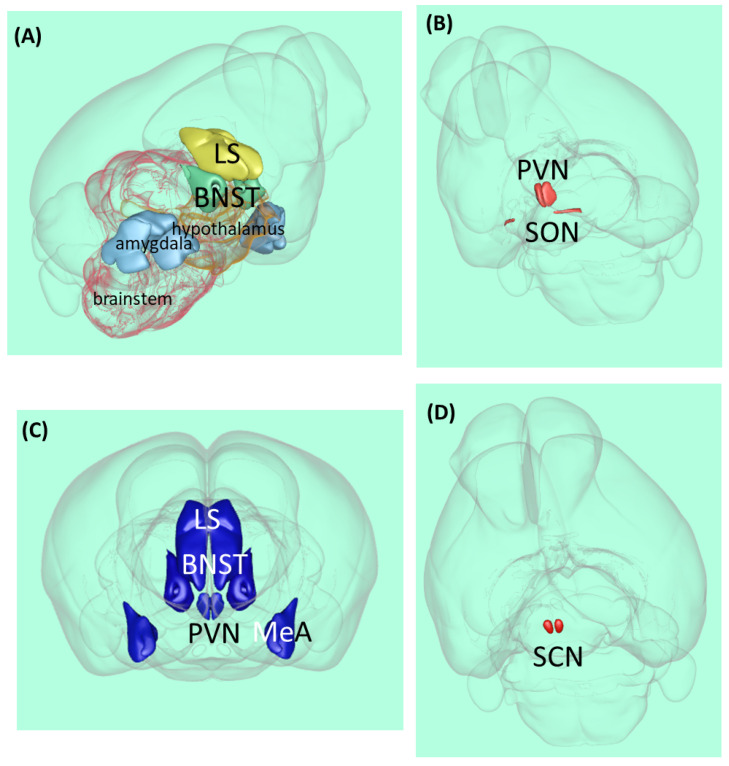
Vasopressin-related brain areas in the rodent brain. (**A**) Localization of the major V1a receptor-containing areas. (**B**) Magnocellular vasopressinergic cells. (**C**) Major parvocellular vasopressinergic cells. (**D**) Suprachiasmatic nucleus. Abbreviations: BNST—bed nucleus of stria terminalis, LS—lateral septum, MeA—medial amygdala, PVN—paraventricular hypothalami nucleus, SCN—suprachiasmatic nucleus, and SON—supraopticus nucleus [29].

**Figure 2 ijms-23-15467-f002:**
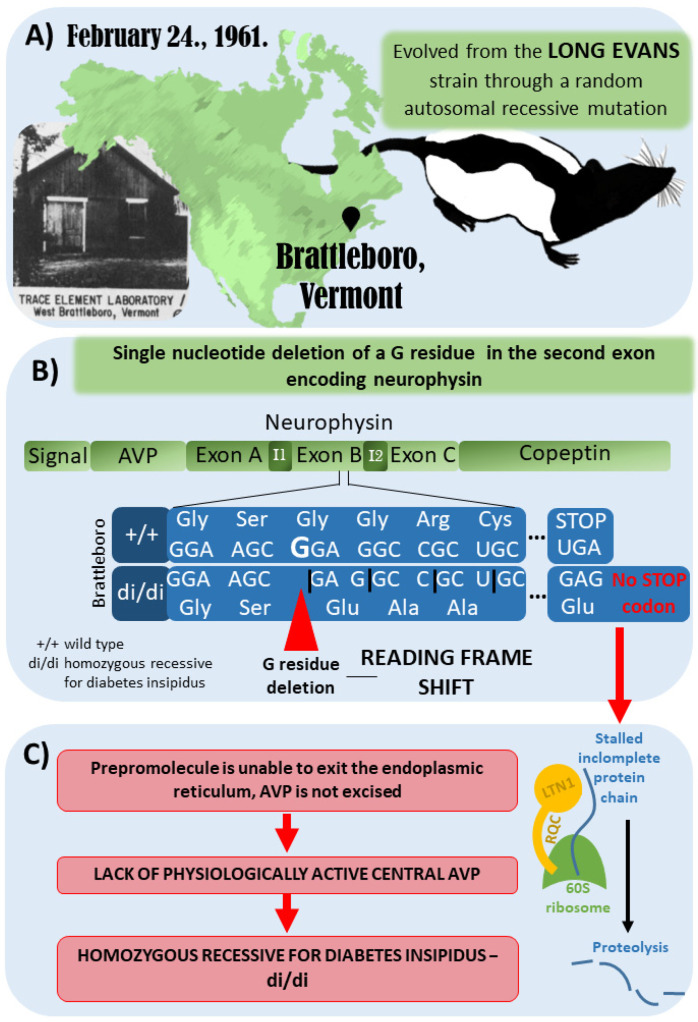
Development of the Brattleboro rat strain. (**A**) Brattleboro rat strain was discovered in 1961 in Brattleboro, Vermont. It evolved from the Long–Evans rat strain through a random autosomal recessive mutation. (**B**) The genetic mutation of Brattleboro strain is a single nucleotide deletion of a G residue in the second exon of neurophysin gene. Due to this mutation, a reading frame shift develops, which results in different C-terminus for the precursor hormone. (**C**) Due to the lack of stop codon, the mRNA cannot leave the ribosomes after translation, and the incomplete protein-chain might also stall in the ribosome. Therefore, a ubiquitin ligase (LTN1), which is part of the ribosome-associated quality control complex, will induce proteolysis. All in all, vasopressin (AVP) is missing; thus, there is no physiologically active central AVP, leading to central diabetes insipidus with polydipsia and polyuria [41,42,43,44,45].

**Figure 4 ijms-23-15467-f004:**
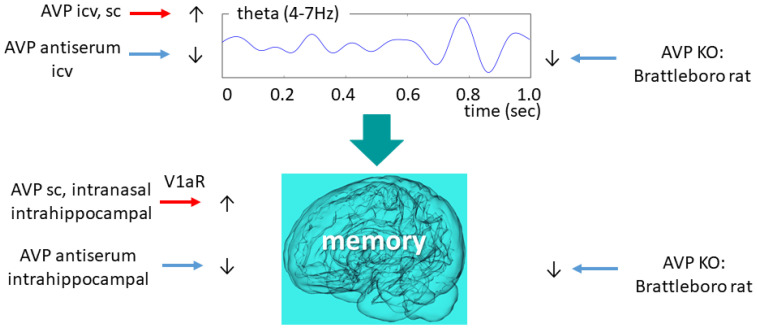
Vasopressinergic influence on theta oscillation and memory formation. Similar intervention induces similar changes in both highly connected measures. AVP—arginine vasopressin; ic—intracerebroventricular; KO—knockout; V1aR—vasopressin 1a type receptor.

## Data Availability

Not applicable.

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
