# Peer review of "Vasopressin as a Possible Link between Sleep-Disturbances and Memory Problems"

_ijms, 2022, doi:10.3390/ijms232415467_

Round 1
Reviewer 1 Report
This review analyses the current evidence on the role of vasopressin (AVP) on sleep and memory and their interaction. In particular the authors describe the contribution of the AVP deficient Battleboro rats to the elucidation of the role of AVP in brain function.
Overall this manuscript provides a useful review of the current knowledge of AVP in relation to sleep and memory.
The manuscript could be improved by explicitly stating in the abstract that this work is indeed a review article.
It would also be useful for a native English speaker to improve some of the sentences in the review
Listed below are some recommended minor changes
line 21: change to '...origin WERE implicated....'
line 28: change to ' Better understanding of the mechanism of...'
line 40 : change to '...disturbances have already been described'
line 47: change to '..latter...'
line 54: remove 'more'
line 67: change to '... in the amygdala...'
line 69 change to '.. were detected..'
line 81 change to 'The vast majority...
line 141-143: sentence could be improved to: The Brattleboro strain was developed from a Long Evans colony in 1961, and since then, the two strains may have diverged genetically.
line 171: change to: ' ...depression, it was also...'
line 173: change to : ' Additionally, it has been found that...
line 190: change to : 'Living organisms sleep in order to recover' or remove this short statement
Line 191: remove the word 'All'
Line 232 remove 'On the other hand'
Line 264 Reference [97] should be after '...sleep disorder'
line 268 Change to 'Few studies...'
line 289 Change to '..reduced..'
line 307 Change to 'Even one night..'
line 317 Change to ' Recent findings have challenged...'
line 325 Change to 'This role was first described...'
line 326 Change to ' Similarly, enhanced SON AVP ...'
line 331 Change to "The hippocampus...'
line 343 Change to '.this is not surprising taking into consideraton..'
.
Author Response
Reviewer 1.
This review analyses the current evidence on the role of vasopressin (AVP) on sleep and memory and their interaction. In particular the authors describe the contribution of the AVP deficient Brattleboro rats to the elucidation of the role of AVP in brain function.
Overall this manuscript provides a useful review of the current knowledge of AVP in relation to sleep and memory.
The manuscript could be improved by explicitly stating in the abstract that this work is indeed a review article.
Answer: Thank you very much for the positive evaluation of our work. Now we explained in the abstract that this is a review paper.
It would also be useful for a native English speaker to improve some of the sentences in the review
Answer: We double checked the manuscript to improve English and made all suggested changes listed below.
Listed below are some recommended minor changes
line 21: change to '...origin WERE implicated....'
line 28: change to ' Better understanding of the mechanism of...'
line 40 : change to '...disturbances have already been described'
line 47: change to '..latter...'
line 54: remove 'more'
line 67: change to '... in the amygdala...'
line 69 change to '.. were detected..'
line 81 change to 'The vast majority...
line 141-143: sentence could be improved to: The Brattleboro strain was developed from a Long Evans colony in 1961, and since then, the two strains may have diverged genetically.
line 171: change to: ' ...depression, it was also...'
line 173: change to : ' Additionally, it has been found that...
line 190: change to : 'Living organisms sleep in order to recover' or remove this short statement
Line 191: remove the word 'All'
Line 232 remove 'On the other hand'
Line 264 Reference [97] should be after '...sleep disorder'
line 268 Change to 'Few studies...'
line 289 Change to '..reduced..'
line 307 Change to 'Even one night..'
line 317 Change to ' Recent findings have challenged...'
line 325 Change to 'This role was first described...'
line 326 Change to ' Similarly, enhanced SON AVP ...'
line 331 Change to "The hippocampus...'
line 343 Change to '.this is not surprising taking into consideraton..'
Reviewer 2 Report
The review entitled ‘Vasopressin as a possible link between sleep-disturbances and memory problems’ summarized the vasopressin in regulating sleep in animal model especially in AVP-deficient Brattleboro rat, describing the EEG disturbance and memory deficits. Authors also implied that the AVP administration or its receptors should be regarded as the possible method/target for treatment. The review manuscript confirms the possible connective role of AVP between biological rhythms and memory formation. The manuscript was good in design and written. However, I have a few comments that needs to be addressed as below:
1. Line 44, I understand authors’ intentions to summarize the role of AVP in circadian rhythm using AVP-deficient rat model, but ‘we will focus on finding obtained from the AVP-deficient (homozygous recessive for diabetes insipidus (di/di)) Brattleboro rat’ need more rational sentence instead of just bringing up this sentence.
2. Since the authors focused on the AVP and circadian rhythm, In Line 54-55, more details for circadian rhythm such as sleep-wake regulations should be discussed.
3. Authors composed and described the background and how the Brattleboro rat strain was developed in ‘3. The Brattleboro rat as a model of AVP-deficiency’ which in my point of view should be trimmed but if authors could introduce more animal models such as AVP deficient mice to support the AVP related to circadian rhythm and related phenotypes should be preferred.
4. Line 176-178, authors cited one Neuron paper that ‘AVP, but not VIP-containing neurons (another important neuropeptide of the SCN) are essential for autonomous network synchrony of the SCN’. I believe the conclusion is not fully inclusive. Please see the previous published paper ‘Liu, D., Stowie, A., de Zavalia, N., Leise, T., Pathak, S.S., Drewes, L.R., Davidson, A.J., Amir, S., Sonenberg, N. and Cao, R., 2018. mTOR signaling in VIP neurons regulates circadian clock synchrony and olfaction. Proceedings of the National Academy of Sciences, 115(14), pp.E3296-E3304.’ , which should also be cited to discuss that AVP and VIP are both important in regulating circadian rhythm in SCN.
5. Line 323, ‘7.2. Vasopressin function related to memory’, I can see authors cited paper that use AVP administration to improve memory but since authors just discussed that memory consolidation and AVP is essential participating in sleep-wake cycles, should authors discuss the possibility that vasopressin’s effects on memory is via sleep stages maintains.
Thanks.
Author Response
Reviewer 2.
The manuscript was good in design and written.
Answer: Thank you very much for positive evaluation of our work.
However, I have a few comments that needs to be addressed as below:
- Line 44, I understand authors’ intentions to summarize the role of AVP in circadian rhythm using AVP-deficient rat model, but ‘we will focus on finding obtained from the AVP-deficient (homozygous recessive for diabetes insipidus (di/di)) Brattleboro rat’ need more rational sentence instead of just bringing up this sentence.
Answer: Thank you very much for the suggestion. We corrected the sentences.
- Since the authors focused on the AVP and circadian rhythm, In Line 54-55, more details for circadian rhythm such as sleep-wake regulations should be discussed.
Answer: Thank you very much for the suggestion. The 5.1. subchapter summarized the results on AVP and circadian changes, while 6.3. subchapter summarized the results on AVP and sleep. Many more subchapter was dealing with circadian rhythm as well as sleep in the Brattleboro rats. Now in Line 54-55 we refer to these chapters.
- Authors composed and described the background and how the Brattleboro rat strain was developed in ‘3. The Brattleboro rat as a model of AVP-deficiency’ which in my point of view should be trimmed but if authors could introduce more animal models such as AVP deficient mice to support the AVP related to circadian rhythm and related phenotypes should be preferred.
Answer: Unfortunately, AVP KO mice is not viable, therefore such results are not available.
See: https://www.mousephenotype.org/data/charts?accession=MGI:88121¶meter_stable_id=IMPC_VIA_001_001¶meter_stable_id=IMPC_VIA_063_001¶meter_stable_id=IMPC_VIA_064_001¶meter_stable_id=IMPC_VIA_065_001¶meter_stable_id=IMPC_VIA_066_001¶meter_stable_id=IMPC_VIA_067_001
AVP-Cre mice are available and was used for studying circadian clock, as we also mentioned (DOI: 10.1016/j.yhbeh.2020.104888 ). However, V1a and V1b receptor KO mice are available. We included some more data on them.
- Line 176-178, authors cited one Neuron paper that ‘AVP, but not VIP-containing neurons (another important neuropeptide of the SCN) are essential for autonomous network synchrony of the SCN’. I believe the conclusion is not fully inclusive. Please see the previous published paper ‘Liu, D., Stowie, A., de Zavalia, N., Leise, T., Pathak, S.S., Drewes, L.R., Davidson, A.J., Amir, S., Sonenberg, N. and Cao, R., 2018. mTOR signaling in VIP neurons regulates circadian clock synchrony and olfaction. Proceedings of the National Academy of Sciences, 115(14), pp.E3296-E3304.’ , which should also be cited to discuss that AVP and VIP are both important in regulating circadian rhythm in SCN.
Answer: Thank you very much for drawing our attention to this paper. We extended our discussion with this and other similar references.
- Line 323, ‘7.2. Vasopressin function related to memory’, I can see authors cited paper that use AVP administration to improve memory but since authors just discussed that memory consolidation and AVP is essential participating in sleep-wake cycles, should authors discuss the possibility that vasopressin’s effects on memory is via sleep stages maintains.
Answer: We incorporated this idea into the discussion.